# Effectiveness of Bivalent Omicron-Containing Booster Vaccines against SARS-CoV-2 Omicron Variant among Individuals with and without Prior SARS-CoV-2 Infection

**DOI:** 10.3390/v15081756

**Published:** 2023-08-17

**Authors:** Kristin Widyasari, Jieun Jang, Taejoon Kang, Sunjoo Kim

**Affiliations:** 1Gyeongsang Institute of Health Sciences, Gyeongsang National University, Jinju 52727, Republic of Korea; kristinwidyasari@gmail.com; 2Gyeongnam Center for Infectious Disease Control and Prevention, Changwon 51154, Republic of Korea; jjegncdc@gmail.com; 3Bionanotechnology Research Center, Korea Research Institute of Bioscience and Biotechnology (KRIBB), Daejon 34141, Republic of Korea; kangtaejoon@gmail.com; 4School of Pharmacy, Sungkyunkwan University, Suwon 16419, Republic of Korea; 5Department of Laboratory Medicine, College of Medicine Gyeongsang National University, Jinju 52727, Republic of Korea; 6Department of Laboratory Medicine, Gyeongsang National University Changwon Hospital, Changwon 51472, Republic of Korea

**Keywords:** SARS-CoV-2, COVID-19, bivalent mRNA vaccine, neutralizing antibody, Omicron variant

## Abstract

In this study, we evaluated the effectiveness of the bivalent mRNA COVID-19 vaccines against the Omicron variant in individuals with or without prior SARS-CoV-2 infection history. We assessed the SARS-CoV-2-specific neutralizing antibody in serum samples by surrogate virus neutralizing assay (sVNT) and determined the serum’s neutralizing capacity against the Omicron BA.5 by a plaque reduction neutralizing test (PRNT50). The results of the sVNT assay demonstrate a higher percentage of inhibition of the serum samples from the infected group than from the uninfected group (*p* = 0.01) before the bivalent vaccination but a similarly high percentage of inhibition after the vaccination. Furthermore, the results of the PRNT50 assay demonstrate a higher neutralizing capacity of the serum samples against Omicron BA.5 in the infected group compared to the uninfected group, both before and after the bivalent vaccine administration (*p* < 0.01 and *p* = 0.02 for samples collected before and after the bivalent vaccination, respectively). A higher neutralizing capacity of the serum samples against BA.5 following bivalent vaccination compared to those before vaccination suggests the efficacy of bivalent mRNA COVID-19 vaccines in triggering an immune response against the Omicron variant, particularly BA.5, regardless of infection history.

## 1. Introduction

The initial coronavirus disease 2019 (COVID-19) vaccines have shown safety and effectiveness against the first wave of severe acute respiratory syndrome coronavirus 2 (SARS-CoV-2) [1]. These initial vaccines are mostly based on the transient expression of the viral spike (S) protein derived from the ancestral strain, the Wuhan strain, to induce specific immune responses to SARS-CoV-2 [2]. Any mutations that occur in the S protein, however, may increase viral resistance to neutralizing antibodies and are associated with a reduction in vaccine effectiveness [3,4]. Hence, the emergence of multiple new variants of SARS-CoV-2 with several mutations in their spike proteins may affect the effectiveness of the initial COVID-19 vaccines.

The initial COVID-19 vaccines were reported to be effective against the Wuhan strain and Alpha variant, but moderately to less effective against the Beta, Gamma, and Delta variants [5,6]. The enhanced transmissibility of the highly contagious Delta variant, which was first discovered in India and later became predominant worldwide from late 2020 to late 2021, has been associated with critical mutations in the S protein, such as D614G, L452R, P681R, and T478K [7]. The L452R mutation, located in the receptor-binding domain (RBD) in immediate proximity to hACE2, has been associated with stronger affinity and immune escape, thus more likely increasing viral transmissibility and pathogenicity [8].

By December 2021, the Omicron variant was identified, which spread rapidly around the world within just a few weeks, thereby replacing the Delta variant as the main variant of concern [9]. Notably, the Omicron variant is not a single strain but has evolved into several subvariants. The first three subvariants of the Omicron variant are BA.1, BA.2, and BA.3. BA.1 was once the most prevalent strain in many countries, replacing the Delta variant [10], but was later replaced by BA.2, and the most recent, subvariants BA.4 and BA.5, emerged and gradually replaced the previous subvariants [11].

Unlike other variants that only harbored several mutations, the Omicron variant was reported to harbor 37 mutations in its spike protein (S), half of which are located in the RBD region [12]. The mutation that occurred in the S protein RBD region may increase the surface positive charge or polarity, resulting in enhanced virus binding affinity to hACE2 and immune evasion [13]. Moreover, to date, many studies have reported evidence of Omicron’s ability to evade the immune response induced by natural infection, therapeutic antibodies, or the initial COVID-19 vaccines [14,15,16], leading to an unprecedented rise in Omicron-related COVID-19 cases.

As SARS-CoV-2 continues to mutate and evolve, multiple new subvariants of Omicron have been detected and reported. The Omicron variant was reported to be less virulent than the preceding variants, such as the Delta variant. However, due to its high transmissibility and ability to evade the immune response, the Omicron variant and its subvariants have become a major concern worldwide [14]. Subsequently, a new countermeasure is needed to rapidly respond to the current circulating variant of concern. The bivalent mRNA COVID-19 vaccines were expected to answer this concern. Unlike the initial COVID-19 vaccines (monovalent), the bivalent mRNA COVID-19 vaccines comprise an mRNA encoding the specific protein of the Wuhan strain and an mRNA encoding the specific protein of the updated Omicron BA.1, BA.4, or BA.5 subvariants [15,16]. In this study, we aimed to evaluate the immunogenicity of the bivalent mRNA COVID-19 vaccines that were administered in South Korea by comparing the level of the SARS-CoV-2-specific neutralizing antibody before and after the bivalent mRNA COVID-19 vaccination. In addition, we also evaluated the neutralizing capacity of the serum samples from the vaccinees with and without a history of SARS-CoV-2 infection against the circulating variant, Omicron BA.5.

## 2. Material and Methods

### 2.1. Study Design

A total of 69 individuals aged 18 years or older, comprising 38 individuals without and 31 individuals with prior SARS-CoV-2 infection, participated in this study. All participants received the initial monovalent COVID-19 vaccines, which were administered in two doses for a completed series. The participants were grouped into uninfected and infected groups. The uninfected group was defined as individuals who had previously received a complete monovalent COVID-19 vaccination and had no infection history or contact with COVID-19-infected individuals. The infected group was defined as individuals who had received a complete monovalent COVID-19 vaccination, had a history of COVID-19 disease, and were recovered for at least four months (median: 8 (4–9) months) before the bivalent mRNA COVID-19 vaccination. The participants with an infection history were confirmed positive for COVID-19 between March and July 2022 and received the bivalent mRNA vaccination between November and December 2022. According to the published studies, the prevalent SARS-CoV-2 variant in Korea between March and July 2022 was the Omicron variant [17,18]; hence, the individuals in the infected group of this study were likely infected by the SARS-CoV-2 Omicron variant. The participants were administered a single dose of the bivalent mRNA COVID-19 vaccine through intramuscular injection. The bivalent mRNA COVID-19 vaccines used in this study were either Moderna BA.1 (BA.1 MD), Pfizer-BioNTech BA.1 (BA.1 PF), or Pfizer-BioNTech BA.4/5 (BA.4/5 PF). The Moderna bivalent mRNA COVID-19 vaccine contains two mRNAs (1:1 ratio, 25 µg each) encoding the prefusion-stabilized spike glycoproteins of SARS-CoV-2 Wuhan strain and the Omicron variants (either BA.1 or BA.4/5). Meanwhile, the Pfizer-BioNTech bivalent mRNA COVID-19 vaccine contains 15 µg of mRNA directed against the SARS-CoV-2 Wuhan strain and 15 µg of mRNA directed against the Omicron variant. Participants with a history of specific allergies, pregnant women, or participants receiving immunosuppressants were excluded from the study. All participants agreed and submitted written consent for this clinical study. The study protocol was approved by the institutional review board of Gyeongsang National University Changwon Hospital (IRB No. 2021-06-022).

### 2.2. Participant Recruitment and Sample Collection

The participant recruitment and the first blood sample collection (one day before vaccination) were conducted from November to December 2022. Subsequently, the second sample collection was conducted from February to March 2023; between two and three months after the bivalent vaccine administration, and the assessment was conducted in March 2023. For the analysis of the neutralizing antibodies, 5 mL of whole blood was collected in a serum separation tube (SST) and centrifuged at 2000× *g* for 10 min. The obtained sera were aliquoted and stored at −80 °C before the analysis.

### 2.3. Detection of the SARS-CoV-2-Specific IgM/IgG

The detection of the SARS-CoV-2-specific IgM and IgG was conducted using the STANDARD Q COVID-19 IgM/IgG Plus (SD Biosensor, Suwon, Republic of Korea), which is a rapid chromatographic immunoassay for the qualitative detection of specific IgM and IgG to SARS-CoV-2 [19]. The test was conducted according to the manufacturer’s instructions. Briefly, 10 µL of serum blood was added to the specimen well of the test device. Subsequently, three drops (90 µL) of buffer were added vertically into the specimen well of the test device. The test result was read after 10–15 min. Colored bands that appeared in all of the control (C), IgM (M), and/or IgG (G) lines demonstrated a positive test result, and a colored band that appeared only in the C line demonstrated a negative test result; meanwhile, if a colored band appeared only in the test lines (M and/or G), or if none of the colored bands appeared, the test was invalid [20].

### 2.4. Neutralizing Antibody Assay

The NAbs in the sera were detected using a cPass SARS-CoV-2 surrogate virus neutralizing antibody test kit (GenScript, Piscataway, NJ, USA), hereafter referred to as the cPass sVNT. The cPass sVNT is an ELISA-based surrogate neutralization assay. This assay uses the basic interaction between the purified protein component of the SARS-CoV-2 receptor-binding domain (RBD) and human angiotensin-converting enzyme 2 (ACE2) in a competitive ELISA-based platform [21,22].

To assess the NAbs, the cPass sVNT kit was used according to the manufacturer’s instructions. In brief, the sera of the participants were mixed with dilution buffer at a ratio of 1:10. The horseradish peroxide-conjugated recombinant SARS-CoV-2-RBD (HRP-RBD) solution was added and incubated at 37 °C for 30 min. The mixtures were subsequently incubated for 15 min at 37 °C in plates that had been precoated with the ACE2 protein. After washing, tetramethyl benzidine substrate solution (TMB) was added and incubated in the dark at 20 °C for 15 min. A stop solution was then added to halt the reaction, and the absorbance was read at 450 nm on an ELISA plate reader. The results were interpreted as positive according to the manufacturer’s recommendations when the inhibition value was ≥ the cutoff value (30%), indicating the presence of an anti-SARS-CoV-2 NAb.

### 2.5. Plaque Reduction Neutralization Test (PRNT)

The neutralizing activity analysis of the serum samples was conducted by a 50% plaque reduction test (PRNT_50_) against the Omicron BA.5 subvariant. The obtained sera were aliquoted and sent to an external institution (Korea University, College of Medicine Department of Microbiology/Institute for Viral Diseases, Seoul, Republic of Korea) for the PRNT assay. To determine the neutralizing activity of the serum samples, the plaque reduction neutralization test (PRNT) was conducted using a SARS-CoV-2 strain isolated in Korea in 2022 (hCoV-19/Korea/KDCA17739542/2022, National Culture Collection for Pathogens (NCCP 43426)). The isolated strain was also used for the neutralization activity analysis using the plaque reduction neutralization test (PRNT). Serum samples were serially diluted twofold and mixed with an equal volume of virus solution containing about 100 plaque-forming units of the virus, followed by incubation at 37 °C for 1 h. Subsequently, the whole reaction mixture was used to inoculate Vero cells for PRNT. Virus-inoculated Vero cell plates were overlaid with agar and incubated at 37 °C for 3 days, followed by staining and plaque counting to measure the PRNT_50_. The PRNT_50_ titer was calculated as the highest serum dilution, which showed a 50% reduction in the number of viral plaques compared to that of a PBS-treated control. The 50% neutralization dose (ND50) of each sample was determined by the Spearman–Karber method.

### 2.6. Statistical Analysis

The differences in the percentages of inhibition from the cPass sVNT and 50% neutralization dose (ND50) from the PRNT_50_ assay in the uninfected and infected groups were assessed based on a repeated measures analysis of variance (RM-ANOVA). All statistical analyses for these assessments were two-tailed tests with a type I error of 5%, and they were performed using SAS software ver. 9.4 (SAS Institute Inc., Cary, NC, USA).

## 3. Results

### 3.1. Cohort of Study

A total of 69 individuals with a median age of 42 (25–63) years old, comprising 69.6% females, participated in this study. Among the total participants, 7.8% and 6.4% received the BA.1 MD bivalent vaccine, 2.6% and 0% received the BA/1 PF vaccine, and 86.5% and 93.6% received the BA.4/5 PF vaccine in the uninfected and infected groups, respectively. One participant (2.6%) from the uninfected group received a bivalent vaccine; however, the participant could not recall which vaccine had been received (Table 1).

### 3.2. The SARS-CoV-2 Specific Antibodies Were Detected Regardless of the Infection Status

The detection of the SARS-CoV-2-specific IgM/IgG antibodies in the serum samples from both the uninfected and infected groups demonstrated a high test positivity. The uninfected group demonstrated a positivity of up to 97.4%, with only 1 out of 38 uninfected samples showing a negative result. Meanwhile, the infected group demonstrated a positivity of up to 100% (Table 2).

### 3.3. The Bivalent mRNA COVID-19 Vaccines Boost the Neutralizing Antibody Response

The evaluation of the neutralizing antibody in the serum samples after bivalent mRNA COVID-19 vaccination demonstrated a positivity of up to 100% in both tested groups (Table 3).

Furthermore, our study demonstrated that the bivalent mRNA COVID-19 vaccination triggered a high percentage of inhibition in both groups. In the preliminary experiment, we found that at a 1:10 dilution, inhibition was close to 100% for all sera; hence, comparing the percentage of inhibition among the groups was not feasible. At a 1:20 dilution, the median percentage of inhibition before administering the bivalent vaccine was significantly higher in the infected group than in the uninfected group (*p* = 0.01), with values of 42.8% (25.1–63.4%) and 65.0% (46.1–77.2%) for the uninfected and infected groups, respectively (Figure 1, Table 4).

These values increased significantly after the administration of bivalent vaccines (median 91.9% (71.6%–96.4%), *p* = 0.001, and 90.5% (67.2%–98.1%), *p* < 0.001, for the uninfected and infected groups, respectively) (Figure 1, Table 4). After bivalent vaccination, the responses in the two groups (uninfected vs. infected) were not significantly different (*p* = 0.89).

### 3.4. The Bivalent mRNA COVID-19 Vaccines Induce Neutralizing Capacity against the Omicron Variant

We also evaluated the neutralizing capacity of the serum samples from all participants after bivalent vaccination by PRNT_50_ assay against the Omicron BA.5 subvariant. Consistent with the sVNT assay, the PRNT_50_ demonstrated an increase in the serum’s neutralizing capacity against Omicron BA.5. The 50% neutralization dose (ND50) after bivalent vaccine administration in both the uninfected and infected groups was significantly higher than before vaccination, with medians of 378.5 (180.7–828.1), *p* < 0.001, and 823.6 (370.0–1173.2), *p* < 0.001, for the uninfected and infected groups, respectively (Figure 2, Table 5).

Subsequently, the comparison of the ND50 after bivalent vaccination between the tested groups demonstrated a higher ND50 in the infected group than in the uninfected group (*p* = 0.02) (Figure 2, Table 5). 

## 4. Discussion

In the study described here, SARS-CoV-2-specific antibodies were detected in nearly all serum samples before the administration of bivalent mRNA COVID-19 vaccines, most likely due to the initial vaccination and/or previous SARS-CoV-2 infection. Several studies have reported the possibility of antibody-mediated protection for SARS-CoV-2 following the initial vaccination or infection lasting for 1–2 years [23,24]. Hence, the high positivity of the SARS-CoV-2-specific IgM/IgG detection tests before bivalent vaccination was likely due to the persistent antibodies produced in response to prior infection or the initial monovalent COVID-19 vaccination.

Nevertheless, the persistence of SARS-CoV-2-specific antibodies triggered by the initial monovalent COVID-19 vaccination or previous infection may not provide sufficient protection against the Omicron variant and its subvariants, which have become the predominant variants worldwide. The initial monovalent COVID-19 vaccines that were administered to all participants comprised genetic material encoding specific proteins of the Wuhan strain. Hence, the antibodies triggered by this initial COVID-19 vaccine were likely highly effective against the SARS-CoV-2 Wuhan strain but less effective against other variants of concern, particularly the Omicron variant.

As reported by several studies, the Omicron variant and its subvariants demonstrate a strong ability to escape the neutralizing antibody elicited by the initial COVID-19 vaccination, previous infection, and monoclonal antibodies due to multiple mutations that are present in their spike proteins [25,26,27]. A published study reported that the Omicron variant has about 37 mutations in the spike protein, with 15 mutations among them located in the RBD, which is responsible for interacting with the angiotensin-converting enzyme 2 (ACE2) [28]. Among these mutations, some mutations, such as K417N and N501Y, are very concerning due to their contribution to immune escape and higher infectivity of the Omicron variant [29,30]. Owing to its enhanced transmissibility, Omicron has rapidly become the dominant variant all over the world.

To address this issue, the U.S. Food and Drug Administration (FDA) authorized the emergency use of bivalent mRNA COVID-19, which contains two mRNA components of the SARS-CoV-2 virus (the ancestral strain and common lineages of the Omicron) [31]. Due to the presence of two mRNA components, the bivalent mRNA COVID-19 vaccines are expected to provide broad protection against COVID-19 caused by not only the original strain (Wuhan strain) but also by currently circulating variants, particularly the Omicron variant. In Korea, the bivalent vaccines began to be administered nationwide in October 2022 [32]. According to the guidelines from the Korean Ministry of Health and Welfare, the BA.1-based bivalent vaccines were the first to be administered, followed by the BA.4- and BA.5-based vaccines [33]. In the first week of November 2022, about one month after the official administration of BA.1-based bivalent vaccines, 91% of the total reported SARS-CoV-2 infections in Korea were due to the BA.5 [34]. Thus, during our study period, the administration of BA.4-/BA.5-based bivalent vaccines became the priority in Korea. Hence, a total of 89% of participants in our study received the BA.4-/BA.5-based bivalent vaccines, and only a small portion received the BA.1-based bivalent vaccine.

Analysis of the neutralizing antibody by sVNT demonstrated that the percentage of inhibition of the serum samples collected before the administration of bivalent mRNA COVID-19 vaccines was significantly higher in the infected group than in the uninfected group. Individuals who have been both infected with and vaccinated against SARS-CoV-2 may have a robust immune response compared to those who have only been vaccinated [35]. This difference is likely due to the repeated stimulation of B-cell responses and antibody production in individuals who have been both infected with and vaccinated against SARS-CoV-2 [36]. Subsequently, the sVNT assay of serum samples collected after bivalent mRNA COVID-19 vaccination demonstrated a significant increase in inhibition in both the uninfected and previously infected groups, with no statistically significant difference between the two groups (*p* = 0.89).

These results were further confirmed by the virus neutralization assay through PRNT_50_ against Omicron subvariant BA.5. Before the bivalent vaccine administration, the PRNT_50_ levels against Omicron subvariant BA.5 were concordant with those observed in the sVNT assay, with the ND50 in the infected group significantly higher than in the uninfected group. Following the bivalent vaccine administration, a higher neutralizing capacity against Omicron BA.5 than before vaccination was measured in the PRNT_50_ assay, with the infected group showing a higher neutralizing capacity than the uninfected group. These results suggest that the bivalent mRNA COVID-19 vaccines, particularly the bivalent BA.4/5 vaccine, which was dominantly administered during the study period, were effective against Omicron BA.5.

Although bivalent mRNA COVID-19 vaccines comprise both mRNA components of the SARS-CoV-2 Wuhan and Omicron variants, in this study, we did not assess the neutralizing activity against the Wuhan strain by the PRNT_50_ assay, which is the gold standard test for the detection and quantification of neutralizing antibodies. This may become one of the limitations of our study. However, we reasoned that the conventional assay used in this study to assess the neutralizing antibodies, i.e., the COVID-19 cPASS sVNT assay, was isotype- and variant-independent, and hence, able to detect SARS-CoV-2-specific neutralizing antibodies and measure the percentage of inhibition against SARS-CoV-2, regardless of their variants [21]. Additionally, the main objective of this study was to evaluate the immunogenicity of the bivalent mRNA COVID-19 vaccines against the Omicron variant, which is currently the prevalent variant in Korea. Another limitation of our study was our inability to compare the effectiveness of bivalent and monovalent vaccines because the booster vaccination using monovalent mRNA COVID-19 vaccine was no longer available in Korea during the study period [37]. Nevertheless, our study shows that the administration of the bivalent mRNA COVID-19 vaccines result in a significant increase in the humoral immune response, strongly suggesting protection against the Omicron variant, particularly BA.5.

## 5. Conclusions

The emergence of the Omicron variant, which became the dominant variant in November 2021, has become a new concern worldwide, mainly due to its fast transmissibility and ability to escape the initial vaccine-triggered immune response. In regard to this issue, the administration of the bivalent mRNA COVID-19 vaccines was expected to provide broader protection against the original SARS-CoV-2 and the currently circulating variants.

In our study, we demonstrated that the administered bivalent mRNA COVID-19 vaccines boosted the humoral immune response and neutralizing capacity against Omicron BA.5, particularly in individuals with a history of SARS-CoV-2 infection. Hence, our findings support the importance of administering bivalent mRNA COVID-19 vaccines to reduce Omicron-related hospitalization and mortality.

## Figures and Tables

**Figure 1 viruses-15-01756-f001:**
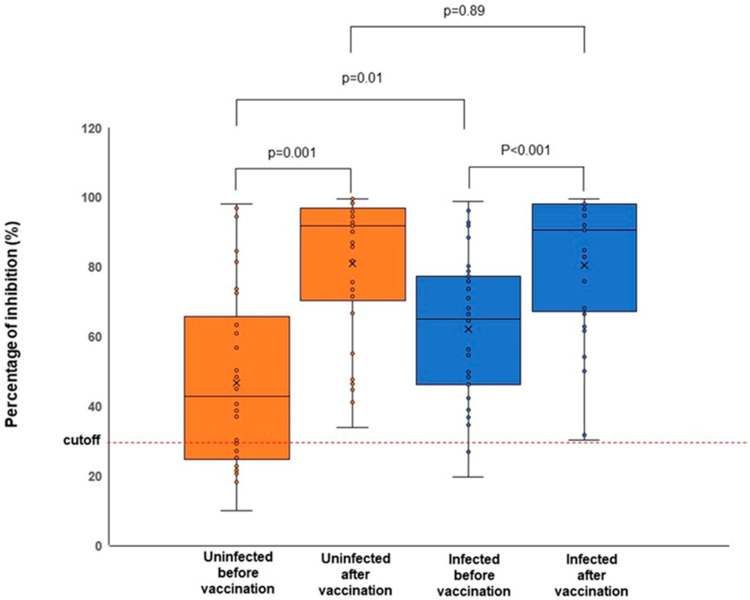
Percentages of inhibition of the sera samples from the uninfected and infected groups before and after the administration of bivalent mRNA COVID-19 vaccines. The sera were diluted up to a ratio of 1:20. Administration of bivalent mRNA COVID-19 vaccines triggered a significant increase in virus inhibition in both groups.

**Figure 2 viruses-15-01756-f002:**
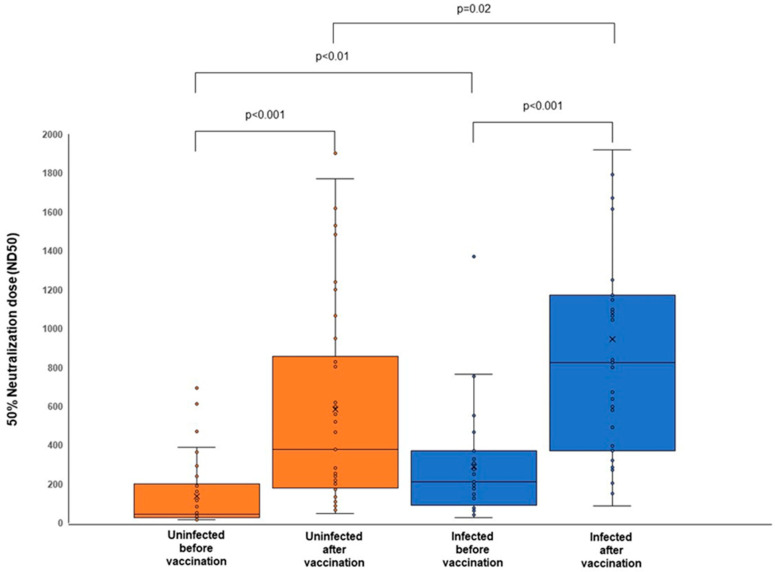
The 50% neutralization dose (ND50) of the serum samples following the administration of bivalent mRNA COVID-19 vaccines against Omicron BA.5. The bivalent vaccine administration triggered a significantly higher neutralizing capacity against the BA.5 compared before vaccination, with the infected group having a higher ND50 than the uninfected group.

**Table 1 viruses-15-01756-t001:** Characteristics of the study subjects.

Characteristics	Uninfected Population N = 38	Infected Population N = 31
Sex		
Male, N (%)	12 (31.6)	9 (29.1)
Female, N (%)	26 (68.4)	22 (70.9)
Age (years), median (range)	43 (28–63)	42 (25–58)
Vaccine type		
BA.1 MD, N (%)	3 (7.8%)	2 (6.4%)
BA.1 PF, N (%)	1 (2.6%)	0 (0%)
BA.4/5 PF, N (%)	33 (86.8%)	29 (93.6%)
N/D, N (%)	1 (2.6%)	0 (0%)

Note: MD, Moderna; PF, Pfizer-BioNTech, N/A, not available; N/D, not defined.

**Table 2 viruses-15-01756-t002:** STANDARD Q COVID-19 IgM/IgG Plus test positivity.

STANDARD Q COVID-19 IgM/IgG Plus	Uninfected (N = 38), %	Infected (N = 31), %
Negative	1 (2.6%)	0 (0%)
Positive	37 (97.4%)	31 (100%)

**Table 3 viruses-15-01756-t003:** GenScript cPASS SARS-CoV-2 sVNT test positivity.

cPass sVNT	Uninfected (N = 38), %	Infected (N = 31), %
Negative	0 (0%)	0 (0%)
Positive	38 (100%)	31 (100%)

**Table 4 viruses-15-01756-t004:** Percentages of the neutralizing antibody level in the uninfected and infected groups following the administration of bivalent mRNA vaccines.

Vaccination Status	Uninfected (N = 38)		Infected (N = 31)	
	N	Median (IQR)	N	Median (IQR)
Before vaccination	38	42.8% (25.1–63.4%)	31	65.0% (46.1–77.2%)
After vaccination	38	91.9% (71.6–96.4%)	31	90.5% (67.2–98.1%)
BA.1 MD	3	90.1% (86.5–99.6%)	2	96.5% (94.8–98.1%)
BA.1 PF	1	96.0% (96.0–96.0%)	0	N/A
BA.4/5 PF	33	91.8% (66.7–96.0%)	29	84.9% (67.2–96.8%)
N/D	1	98.3% (98.3–98.3%)	0	N/A

Note: MD, Moderna; PF, Pfizer-BioNTech, N/A, not available; N/D, not defined.

**Table 5 viruses-15-01756-t005:** The 50% neutralization dose (ND50) of the serum samples against Omicron BA.5.

Vaccination Status	Uninfected (N = 38)		Infected (N = 31)	
	N	Median (IQR)	N	Median (IQR)
Before vaccination	38	42.3 (26.2–199.3)	31	209.5 (91.5–369.3)
After vaccination	38	378.5 (180.7–828.1)	31	823.6 (370.0–1173.2)
BA.1 MD	3	215.5 (46.4–1617.6)	2	1088.1 (1083.1–1093.1)
BA.1 PF	1	565.7 (565.7–565.7)	0	N/A
BA.4/5 PF	33	377.7 (180.7–803.5)	29	800.8 (370.0–1173.2)
N/D	1	828.1 (828.1–828.1)	0	N/A

Note: MD, Moderna; PF, Pfizer-BioNTech, N/A, not available; N/D, not defined.

## Data Availability

All data underlying the results are available as part of the article and no additional source data are applicable.

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
