# Peer review of "Effectiveness of Bivalent Omicron-Containing Booster Vaccines against SARS-CoV-2 Omicron Variant among Individuals with and without Prior SARS-CoV-2 Infection"

_viruses, 2023, doi:10.3390/v15081756_

Round 1
Reviewer 1 Report
The authors have investigated the efficacy of bivalent vaccines against Omicron strains, including differences with and without COVID-19 infection, providing interesting real-world findings. The following points need to be considered. There is no information on how many times the vaccine has been administered before this vaccination. For infected individuals, what was the strain of SARS-COV-2 with which they were infected? At least information about the strains that were prevalent during the period covered by this study is needed. With regard to IgG/IgM antibody titres and neutralising antibody titres, are the antibodies being measured against the Wuhan strain of SARS-COV-2? Clarification on whether these antibody titres correlate with the ND50 against omicron would provide information on whether conventional assays are useful.Author Response
The authors would like to thank the Reviewers and editor for their specific and helpful comments on the manuscript. The authors have carefully considered the comments and have made revisions to the manuscript to address reviewers’ and editors’ concerns.
Responses to the Reviewer are reported as follows:
Reviewer 1:
The authors have investigated the efficacy of bivalent vaccines against Omicron strains, including differences with and without COVID-19 infection, providing interesting real-world findings. The following points need to be considered.
- There is no information on how many times the vaccine has been administered before this vaccination.
Response:
The bivalent mRNA COVID-19 vaccine was administered to individuals that have received completed initial monovalent COVID-19 vaccines. The initial monovalent COVID-19 vaccines require 2 doses for a completed series (two shots). Subsequently, as suggested by the reviewer, we added this information in the methods section of our manuscript.
Page 2, lines: 84-86.
“All participants received the initial monovalent COVID-19 vaccines which were administered in two doses for a completed series.”
- For infected individuals, what was the strain of SARS-COV-2 with which they were infected? At least information about the strains that were prevalent during the period covered by this study is needed.
Response:
As suggested by the reviewer, by considering the prevalent strain during the study period, we have added information on the strain of SARS-CoV-2 which may infect the “infected individuals”
Page 2, lines: 94-97.
“According to the published studies, the prevalent SARS-CoV-2 variant in Korea between March and July 2022 was the Omicron variant [15, 16]; hence, the individuals in the infected group of this study were likely infected by the SARS-CoV-2 Omicron variant.”
- With regard to IgG/IgM antibody titres and neutralising antibody titres, are the antibodies being measured against the Wuhan strain of SARS-COV-2? Clarification on whether these antibody titres correlate with the ND50 against omicron would provide information on whether conventional assays are useful.
Response:
In our study, we used the IgG/IgM test kit to detect the presence of the SARS-CoV-2 specific IgG/IgM antibody before the administration of bivalent mRNA COVID-19 vaccines. This kit detected the antibodies qualitatively. A high positivity of the IgG/IgM test demonstrated the presence of SARS-CoV-2-specific antibodies, which were likely produced in response to the initial monovalent COVID-19 vaccines which comprise material genetic encoding specific protein of the Wuhan strain.
Subsequently, the sVNT assay demonstrated a high positivity, suggesting the presence of the SARS-CoV-2 specific neutralizing antibodies, and a high percentage of inhibition in samples that were collected both before and after the bivalent vaccination. Although we did not test the serum against the Wuhan strains, however, considering the high positivity of both the IgG/IgM test and sVNT test we believed that these conventional assays are highly sensitive for the assessment of the SARS-CoV-2 specific antibodies from the serum samples.
Additionally, the PRNT50 assay of the serum samples collected after the bivalent vaccination demonstrated a high ND50 against Omicron, whereas, this result is in concordance with the sVNT assay where the percentage of inhibition was also observed to increase significantly after the bivalent vaccination.
Together, our results demonstrate that the conventional assay is still useful for the analysis of the SARS-CoV-2 specific antibodies, regardless of the variant of SARS-CoV-2 that is being assessed.
Lines: 290-293, Lines: 303-307
Reviewer 2 Report
Widyasari and colleagues describe immune responses after vaccination with bivalent COVID-19 vaccines in a cohort of participants. As expected, vaccination with bivalent vaccines increases the neutralizing antibody levels against the Omicron variant. Differences were observed between participants that had experiences an earlier SARS-CoV-2 infection and those who did not. The authors conclude that the results show the benefit of vaccination with bivalent vaccines, with the induced immune responses likely to give protection against infection.
The purpose of the study, the results and conclusions are clear. I have a few suggestions for text edits:
Increments are often fixed amounts and in mathematics can even be decreases. Therefore, throughout the document replace ‘increment’ by ‘increase’.
Line 72: change ‘effectiveness’ to ‘immunogenicity’
Lines 80-81 and 162-165: Suggestion to mention receipt of complete monovalent vaccination in the Study Design section and delete in the results section.
Lines 167-170: Delete: ‘The participant recruitment and the first sample collection were conducted from November to December 2022. Subsequently, the second sample collection was conducted from February to March 2023, and the assessment was conducted in March 2023.’ as this is already mentioned in the Material and Methods section.
Line 180-181: Suggestion to change to ‘and with only 1 out of 38 uninfected 180 samples showed showing a negative result.’
In addition, some of the sentences or sections are a bit wordy. Here are some suggestions to shorten the text:
Lines 51-53: ‘By December 2021, the Omicron variant was identified, which spread rapidly around the world within just a few weeks, thereby replacing the Delta variant as the main variant of concern [9].’
Line 185: ‘In our study, we found that the Evaluation of the neutralizing antibody in the serum 185 samples from the uninfected and…. Etc.’
Line 189: ‘Furthermore, our study demonstrated that the administration of bivalent COVID-19 vaccines triggered…. Etc.’
Lines 191-195: ‘At a 1:10 dilution, inhibition was close to 100% for all sera; hence, comparing the percentage of inhibition among the groups was not feasible. At a 1:20 dilution, the median percentage of inhibition before administering the bivalent vaccine was significantly higher… etc.’
Lines 206-209: ‘After bivalent vaccination the responses in the two groups (uninfected vs. infected) were not significantly different (p=0.89) (Table 3).’
Lines 234-236: ‘In the study described here, SARS-CoV-2-specific antibodies were detected in nearly all serum samples before administration of bivalent mRNA COVID-19 vaccines, most likely due to the initial vaccination and/or previous SARS-CoV-2 infection.’
Lines 260-277: ‘Analysis of the neutralizing antibody by sVNT demonstrated that the percentage of inhibition of serum samples collected before the administration of bivalent mRNA COVID-19 vaccines was significantly higher in the infected group than in the uninfected group. Individuals who have been both infected with and vaccinated against SARS-CoV-2. This difference is likely due to the repeated stimulation of B-cell responses and antibody production in individuals who have been both infected with and vaccinated against SARS-CoV-2. Subsequently, the sVNT assay of serum samples collected after bivalent mRNA COVID-19 vaccination demonstrated a significant increase in inhibition in both uninfected and previously infected groups, with no statistically significant difference between the two groups (p=0.89).’
Lines 279-288: ‘Before the bivalent vaccine administration, the PRNT50 levels against Omicron subvariant BA.5 were concordant with those observed in the sVNT assay, with the ND50 in the infected group was significantly higher than in the uninfected group. Following the bivalent vaccine administration, higher neutralizing capacity against Omicron BA.5 than before vaccination was measured in the PRNT50 assay, with the infected group showing a higher neutralizing capacity than the uninfected group. These results suggest that the bivalent mRNA COVID-19 vaccines, particularly the bivalent BA.4/5 vaccine, which was dominantly administered during the study period, were effective against the Omicron BA.5.’
Lines 290-293: ‘Nevertheless, our study shows that the administration of the bivalent mRNA COVID-19 vaccines result in a significant increment of the humoral immune response, strongly suggesting protection against the Omicron variant.’
Author Response
The authors would like to thank the Reviewers and editor for their specific and helpful comments on the manuscript. The authors have carefully considered the comments and have made revisions to the manuscript to address reviewers’ and editors’ concerns.
Responses to the Reviewer are reported as follows:
Reviewer 2:
Widyasari and colleagues describe immune responses after vaccination with bivalent COVID-19 vaccines in a cohort of participants. As expected, vaccination with bivalent vaccines increases the neutralizing antibody levels against the Omicron variant. Differences were observed between participants that had experiences an earlier SARS-CoV-2 infection and those who did not. The authors conclude that the results show the benefit of vaccination with bivalent vaccines, with the induced immune responses likely to give protection against infection.
The purpose of the study, the results and conclusions are clear. I have a few suggestions for text edits:
- Increments are often fixed amounts and in mathematics can even be decreases. Therefore, throughout the document replace ‘increment’ by ‘increase’.
Response:
As suggested by the reviewer, we have replaced “increment” with “increase” throughout the document.
- Line 72: change ‘effectiveness’ to ‘immunogenicity’
Response: As suggested by the reviewer, we have changed the “effectiveness” to “immunogenicity”
Page 2, line: 75.
- Lines 80-81 and 162-165: Suggestion to mention receipt of complete monovalent vaccination in the Study Design section and delete in the results section.
Response:
As suggested by the reviewer, we have mentioned the receipt of complete monovalent vaccination in the Study Design section and deleted it from the results section.
Page 2, lines: 84-92
“All participants received the initial monovalent COVID-19 vaccines which were administered in two doses for a completed series. The participants were grouped into uninfected and infected groups. The uninfected group was defined as individuals who had previously received a complete monovalent COVID-19 vaccination and had no infection history or contact with COVID-19-infected individuals. The infected group was defined as individuals who had received a complete monovalent COVID-19 vaccination, had a history of COVID-19 disease, and had recovered for at least four months (median: 8 [4–9] months) before the bivalent mRNA COVID-19 vaccination.”
- Lines 167-170: Delete: ‘The participant recruitment and the first sample collection were conducted from November to December 2022. Subsequently, the second sample collection was conducted from February to March 2023, and the assessment was conducted in March 2023.’ as this is already mentioned in the Material and Methods section.
Response:
As suggested by the reviewer, we have deleted the mentioned part from the result section.
- Line 180-181: Suggestion to change to ‘and with only 1 out of 38 uninfected samples showing a negative result.’
Response:
As suggested by the reviewer, we have changed the mentioned sentence to “and with only 1 out of 38 uninfected samples showing a negative result”
Page 4, lines 186-187.
In addition, some of the sentences or sections are a bit wordy. Here are some suggestions to shorten the text:
- Lines 51-53: ‘By December 2021, the Omicron variant was identified, which spread rapidly around the world within just a few weeks, thereby replacing the Delta variant as the main variant of concern [9].’
Response:
We have shortened the text as suggested by the reviewer.
Page 2, lines: 50-52.
“By December 2021, the Omicron variant was identified, which spread rapidly around the world within just a few weeks, thereby replacing the Delta variant as the main variant of concern [9].”
- Line 185: ‘In our study, we found that the Evaluation of the neutralizing antibody in the serum 185 samples from the uninfected and…. Etc.’
Response:
We have shortened the text as suggested by the reviewer.
Page 5, lines: 191-193
“The evaluation of the neutralizing antibody in the serum samples after bivalent mRNA COVID-19 vaccination demonstrated a positivity of up to 100% in both tested groups (Table 3).
- Line 189: ‘Furthermore, our study demonstrated that the administration of bivalent COVID-19 vaccines triggered…. Etc.’
Response:
As suggested by the reviewer, we have shortened the text.
Page 5, lines 195-196.
“Furthermore, our study demonstrated that the bivalent mRNA COVID-19 vaccination triggered a high percentage of inhibition in both groups.”
- Lines 191-195: ‘At a 1:10 dilution, inhibition was close to 100% for all sera; hence, comparing the percentage of inhibition among the groups was not feasible. At a 1:20 dilution, the median percentage of inhibition before administering the bivalent vaccine was significantly higher… etc.’
Response:
As suggested by the reviewer, we have shortened the text.
Page 5, lines: 196-200.
“In the preliminary experiment, we found that at a 1:10 dilution, inhibition was close to 100% for all sera; hence comparing the percentage of inhibition among the groups was not feasible. At a 1:20 dilution, the median percentage of inhibition before administering the bivalent vaccine was significantly higher in the infected group than in the uninfected group”
- Lines 206-209: ‘After bivalent vaccination the responses in the two groups (uninfected vs. infected) were not significantly different (p=0.89) (Table 3).’
Response:
As suggested by the reviewer, we have shortened the text.
Page 6, lines: 210-212.
“After bivalent vaccination, the responses in the two groups (uninfected vs. infected) were not significantly different (p=0.89) (Table 3). “
- Lines 234-236: ‘In the study described here, SARS-CoV-2-specific antibodies were detected in nearly all serum samples before administration of bivalent mRNA COVID-19 vaccines, most likely due to the initial vaccination and/or previous SARS-CoV-2 infection.’
Response:
We have shortened the text as suggested by the reviewer.
Page 7, lines: 237-239.
‘In the study described here, SARS-CoV-2-specific antibodies were detected in nearly all serum samples before the administration of bivalent mRNA COVID-19 vaccines, most likely due to the initial vaccination and/or previous SARS-CoV-2 infection.”
- Lines 260-277: ‘Analysis of the neutralizing antibody by sVNT demonstrated that the percentage of inhibition of serum samples collected before the administration of bivalent mRNA COVID-19 vaccines was significantly higher in the infected group than in the uninfected group. Individuals who have been both infected with and vaccinated against SARS-CoV-2. This difference is likely due to the repeated stimulation of B-cell responses and antibody production in individuals who have been both infected with and vaccinated against SARS-CoV-2. Subsequently, the sVNT assay of serum samples collected after bivalent mRNA COVID-19 vaccination demonstrated a significant increase in inhibition in both uninfected and previously infected groups, with no statistically significant difference between the two groups (p=0.89).’
Response:
We have shortened the text as suggested by the reviewer.
Page 8, lines: 278-288.
“Analysis of the neutralizing antibody by sVNT demonstrated that the percentage of inhibition of serum samples collected before the administration of bivalent mRNA COVID-19 vaccines was significantly higher in the infected group than in the uninfected group. Individuals who have been both infected with and vaccinated against SARS-CoV-2 may have a robust immune response compared to those who have only been vaccinated [28]. This difference is likely due to the repeated stimulation of B-cell responses and antibody production in individuals who have been both infected with and vaccinated against SARS-CoV-2 [29]. Subsequently, the sVNT assay of serum samples collected after bivalent mRNA COVID-19 vaccination demonstrated a significant increase in inhibition in both uninfected and previously infected groups, with no statistically significant difference between the two groups (p=0.89). “
- Lines 279-288: ‘Before the bivalent vaccine administration, the PRNT50 levels against Omicron subvariant BA.5 were concordant with those observed in the sVNT assay, with the ND50 in the infected group was significantly higher than in the uninfected group. Following the bivalent vaccine administration, a higher neutralizing capacity against Omicron BA.5 than before vaccination was measured in the PRNT50 assay, with the infected group showing a higher neutralizing capacity than the uninfected group. These results suggest that the bivalent mRNA COVID-19 vaccines, particularly the bivalent BA.4/5 vaccine, which was dominantly administered during the study period, were effective against the Omicron BA.5.’
Response:
We have shortened the text as suggested by the reviewer.
Pages 8-9, lines: 290-298.
“Before the bivalent vaccine administration, the PRNT50 levels against Omicron subvariant BA.5 were concordant with those observed in the sVNT assay, with the ND50 in the infected group significantly higher than in the uninfected group. Following the bivalent vaccine administration, a higher neutralizing capacity against Omicron BA.5 than before vaccination was measured in the PRNT50 assay, with the infected group showing a higher neutralizing capacity than the uninfected group. These results suggest that the bivalent mRNA COVID-19 vaccines, particularly the bivalent BA.4/5 vaccine, which was dominantly administered during the study period, were effective against the Omicron BA.5.”
- Lines 290-293: ‘Nevertheless, our study shows that the administration of the bivalent mRNA COVID-19 vaccines result in a significant increment of the humoral immune response, strongly suggesting protection against the Omicron variant.’
Response:
We have shortened the text as suggested by the reviewer.
Page 9, lines: 313-315.
“Nevertheless, our study shows that the administration of the bivalent mRNA COVID-19 vaccines result in a significant increase in the humoral immune response, strongly suggesting protection against the Omicron variant, particularly BA.5. “